# Correlation between varus-type knee osteoarthritis severity and hindfoot alignment: Analysis of radiographs in the long-leg weight-bearing anteroposterior view

Yusuke Ozaki 📖*, Ryota Hara, Kensuke Okamura, Hiroaki Kurokawa, Yusuke Inagaki, Munehiro Ogawa 📖, Akira Taniguchi, Yasuhito Tanaka

Department of Orthopaedic Surgery, Nara Medical University, Kashihara, Nara, Japan

* zaki0189med@gmail.com

## Abstract

### Background

In knee osteoarthritis, the subtalar joint undergoes valgus and varus contractions to compensate for deformities in the knee joint. In this cross-sectional study, we investigated the relationship between varus-type knee osteoarthritis severity and hindfoot alignment severity by concurrently assessing varus-type knee osteoarthritis severity and hindfoot alignment using radiographs in the long-leg weight-bearing anteroposterior view.

### Patients and methods

A total of 114 patients with knee osteoarthritis graded Kellgren–Lawrence II or higher (128 knees) and 30 healthy controls (31 knees) underwent long-leg weight-bearing anteroposterior imaging for 1 year. Four angles were measured on radiographs in the long-leg weight-bearing anteroposterior view: the femorotibial angle; tibial calcaneal angle; tibial anterior surface angle; and talocrural joint angle between the tibial plafond and talar dome on weight-bearing. Group comparisons were conducted for each Kellgren–Lawrence classification, which was used to classify the severity of knee osteoarthritis at each measured angle. One-way analysis of variance was used to test the results.

### Results

The mean tibial calcaneal angles were 9.7°, 11.3°, 8.8°, and 9.8° in controls and in patients with Kellgren–Lawrence grades II, III, and IV, respectively ($p < 0.05$). The mean femorotibial angles were 175.6°, 176.8°, 180.3°, and 186.2° in controls and in patients with Kellgren–Lawrence grades II, III, and IV, respectively ($p < 0.05$). On weight-bearing, the tibial anterior surface angle and the talocrural joint angle between the tibial plafond and talar dome varied according to severity level.

**Data availability statement:** All relevant data are within the manuscript.

**Funding:** The author(s) received no specific funding for this work.

**Competing interests:** The authors have declared that no competing interests exist.

## Conclusion

In varus-type knee osteoarthritis cases, defined in accordance with the Kellgren–Lawrence classification, hindfoot alignment leaned toward valgus. As the severity of knee osteoarthritis progressed, the valgus of the hindfoot alignment reduced. While future longitudinal analyses are necessary, these observations indicate both potential compensatory changes and their limitations in varus-type knee osteoarthritis.

## Introduction

In knee osteoarthritis (KOA), the progression of joint destruction correlates with varus or valgus knee deformities [1,2]. Particularly, the subtalar joint in the hindfoot exhibits inverted valgus and varus characteristics with varus/valgus deformities in KOA [3]. Moreover, previous studies have reported that the proportion of unsatisfied patients after total knee arthroplasty for KOA ranged from 11% to 25% [4,5], with potential factors including changes in lower extremity joints beyond knee alignment correction. Preoperative and postoperative lower limb alignment may significantly affect patient outcomes. Postoperative valgus alignment (≥3°) leads to good clinical and functional outcomes, whereas varus alignment (≤−3°) is related to undesirable outcomes. Although osteoarthritis in the lower extremities does not necessarily affect lower extremity alignment alone, it is necessary to consider the relationship between KOA and associated changes in foot morphology when considering changes in lower extremity loading and alignment [6].

Previous reports of compensatory valgus and varus contraction of the subtalar joints with the varus and valgus of KOA have primarily utilized subtalar [7,8] or hip–calcaneus [9] radiography as the radiographic imaging method. However, in clinical settings, long-leg weight-bearing anteroposterior (LL-WB AP) radiography is commonly used to evaluate the effects of lower extremity alignment and loading [10–16]. The LL-WB radiograph is a more reliable method of assessing alignment than computer-assisted navigation, computer tomography, and hindfoot alignment radiography [17,18]. Although subtalar radiography is superior for hindfoot evaluation, LL-WB AP radiography has the advantage of simultaneous imaging and assessing the severity of the entire lower limb, from the hip to the hindfoot and knee joint, which was measured on radiographs in the LL-WB AP view in this study. To the best of our knowledge, no study has examined the association between the severity of KOA and changes in hindfoot alignment. Therefore, in this study, we aimed to determine the relationship between varus-type KOA deformity severity and alignment changes in the subtalar joint using radiographs in the LL-WB AP view.

## Materials and methods

### Study design, setting, and period

This retrospective case-control study included patients diagnosed with varus-type KOA who underwent LL-WB AP imaging at our department between January 2020 and December 2020. Data were accessed for research purposes between

September 1, 2021 and December 31, 2023. The authors did not access personally identifiable participant information during or after data collection.

## Study population and inclusion and exclusion criteria

Patients who underwent surgery on at least one of the hip, knee, or talocrural joints, as well as those who underwent imaging that was not valid for reading, were excluded. The control group comprised healthy limbs of patients with knee disorders, such as trauma, ligament injuries, pain, and soft tissue inflammation, whose radiographs were concurrently acquired in the LL-WB AP view. Both knees were evaluated for each patient.

The LL-WB AP view technique is shown in Fig 1. The patients were placed on the ground under a bipedal standing load. The X-ray tube position was set at a distance of 2 m from the patella according to the patellar elevation. Radiographs were captured from the fourth lumbar vertebra to the lower end of the calcaneus [19]. Digital radiographs were used for reading in all cases. One LL-WB AP imaging was performed per patient between January 2020 and December 2020.

## Measures

Four angles were measured to evaluate lower limb alignment. First, the femorotibial angle (FTA) was defined as the lateral angle of the femur and tibia diaphyseal lines. This measurement was conducted using the midpoint of the transverse diameter located 10 cm from the center of the femoral and tibial epiphyses, respectively (Fig 2a). Second, the tibial calcaneal angle (TCA) was defined as the angle between the tibial axis and the line connecting the calcaneal contour contact point, which is tangential to the line perpendicular to the tibial axis and center of the tibial canal. The calcaneal contour was defined as the easily discernible cortical bone contour line of the calcaneus in the LL-WB AP view (Fig 2b). The third angle, known as the tibial anterior surface angle (TAS), was measured as the medial angle between the tibial axis and tibial canopy in the frontal view of the talocrural joint (Fig 2c). Finally, the angle between the tibial plafond and talar dome during weight-bearing (TTW) was determined. This angle was defined as the positive angle for varus deformation of the talocrural joint (Fig 2d).

We defined each angle as above, using the Kellgren–Lawrence (K–L) classification and four measured angles, and analyzed the differences between each group for each severity of varus-type KOA at each measured angle.

Two experienced orthopedic surgeons participated in a reliability test regarding the reliability of the assessment when reading the radiographs. Each orthopedic surgeon used randomly selected radiographs and measured the four angles according to established definitions.

When the intra- and inter-examiner reliabilities were examined, the intraclass correlation coefficients for reliability were as follows: FTA, 0.99 for intra-rater and 0.99 for inter-rater; TCA, 0.97 for intra-rater and 0.97 for inter-rater; TAS, 0.71 for intra-rater and 0.86 for inter-rater; and TTW, 0.99 for intra-rater and 0.95 for inter-rater.

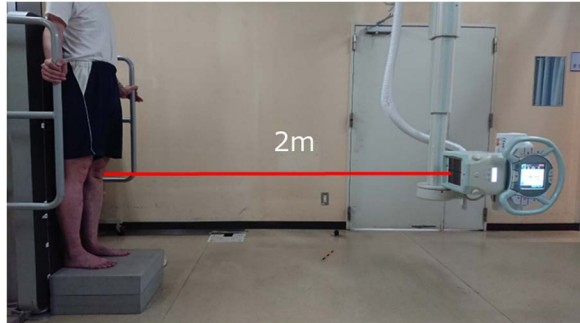

**Fig 1. Method for obtaining long-leg weight-bearing anteroposterior radiographs.** Imaging: Fourth lumbar vertebrae to the lower end of the calcaneus; Middle point of imaging: Patella.

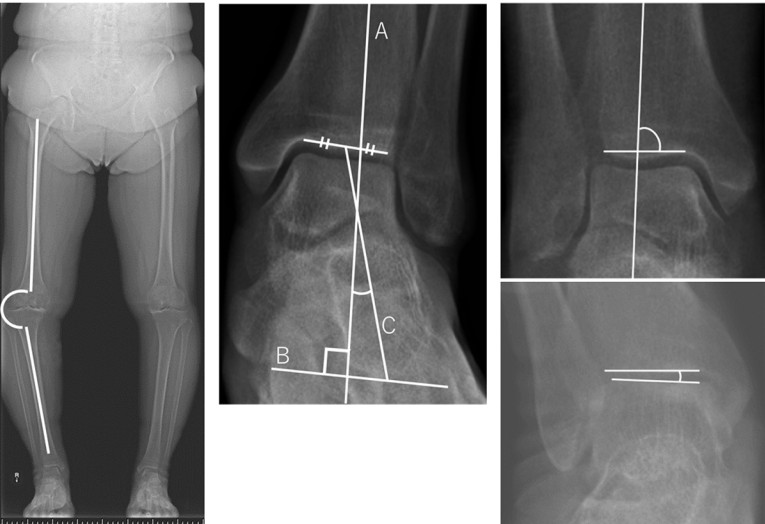

**Fig 2.  a.** Femoral Angle. Lateral angle of intersection of the two straight lines of the femoral and tibial diaphysis. The midpoints of the transverse diameters of the femur and tibia were 10 cm from the center of the epiphyses of the femur and tibia. **b. Tibial Calcaneal Angle.** The angle between the tibial axis and the line tangent to the line perpendicular to the tibial axis was the angle between the calcaneal contour contact point and the straight line connecting the center of the tibial canal. Line A: Tibial anatomical axis; Line B: Perpendicular to the tibial axis; Line C: Tibial canal center—calcaneal contour contact point. **c. Tibial Anterior Surface Angle.** The medial angle between the tibial axis and tibial canopy in the frontal view of the talofemoral joint. **d. Tibial Plafonds and Talar Domes on Weight Bearing.** The angle between the tibiotalar and talar glenoid planes.

Varus-type KOA severity was classified using the K–L classification [20], whereas ankle osteoarthritis severity was assessed using the Takakura–Tanaka (T–T) classification [21,22].

Distinguishing between K–L grade 0 and I KOA based solely on radiographs was challenging. This resulted in the index group being split into three groups: KL grade II, III, and IV.

### Compliance with ethical standards

All procedures performed in the study involving human participants were reviewed and approved by the Nara Medical University Ethics Committee and adhered to the ethical standards of the institution where the study was conducted (Approval No. 3094), the 1964 Declaration of Helsinki, and subsequent amendments or equivalent. Consent was obtained from patients through an open-written opt-out method.

### Statistical analyses

One-way analysis of variance (ANOVA) was performed to assess the significance of the differences between the grade and control groups in the K–L classification across the four measured angles. Additionally, ANOVA was used to assess differences between the stage and control groups in the T–T classification at the TCA. Post-hoc analysis was performed using the Bonferroni test for multiple comparisons. The level of significance was set at $p < 0.05$. All statistical analyses were performed using IBM SPSS software (version 28; Armonk, NY, USA).

## Results

### Participant selection

From January and December 2020, 200 patients with varus-type KOA (216 knees) underwent LL-WB AP imaging. After excluding cases wherein surgery involved more than one joint in the lower limbs, those wherein interpretation of bone

contour shadows was difficult when reading radiographs, and those with insufficiently captured radiographs, the study cohort comprised 114 patients (128 knees).

## Baseline participant characteristics

The distribution of K–L classification within the study cohort was as follows: 35 knees from 28 patients classified as grade II, 55 knees from 52 patients classified as grade III, and 38 knees from 34 patients classified as grade IV. The control group consisted of 31 knees from 30 patients (Table 1).

## KOA severity

The FTA, TCA, TAS, and TTW for each grade group of the K–L classification and control groups are presented in Table 2. Significant differences were observed in the FTA and TCA among the four groups (p<0.05) (Fig 3a, b). Specifically, the mean TCAs were 9.7° for controls, 11.3° for those with grade II, 8.8° for those with grade III, and 9.8° for those with grade IV, with a significant difference observed when comparing those with grade II and those with grade III (p=0.026). No significant difference was observed in the TAS and TTW among the four groups (p=0.94 and p=0.17, respectively) (Fig 3c, d).

## KOA and ankle OA severity

Across all severities of varus-type KOA, ankle osteoarthritis was found to be in its early stages in approximately half of the cases, whereas all talocrural joints in the control group were categorized as stage 0 (Table 3).

## Discussion

This study compared the severity of each varus-type KOA grade with lower limb and hindfoot alignment. Our findings revealed a correlation between the severity of varus-type KOA and FTA, consistent with those of previous reports [1,23,24]. Notably, this study contributes to the literature by investigating the association between varus-type KOA severity and hindfoot alignment. Although previous studies have examined knee and hindfoot alignment, such as in the study by Norton et al. [3], in this work, we examined the association between the severity of varus-type KOA [20] and hindfoot

**Table 1. Baseline characteristics of the patient and control groups.**

|  | Total KOA | K-L grade II | K-L grade III | K-L grade IV | Control |
|---|---|---|---|---|---|
| Person/knee | 114/128 | 28/35 | 52/55 | 34/38 | 30/31 |
| Person (female) | 114(68) | 28(12) | 52(27) | 34(25) | 30(14) |
| Age (y) | 68.4(±12.3) | 61.8(±14.6) | 68.7(±11.6) | 73.3(±8.2) | 39.0(±8.8) |
| BMI (kg/m²) | 24.8(±3.9) | 23.8(±4.0) | 24.7(±3.7) | 25.7(±3.9) | 23.6(±3.4) |

Data are presented as number, number (%), or mean±standard deviation.

**Table 2. Measurement of the four angles and grade progression in the Kellgren–Lawrence classification.**

|  | K-L grade II | K-L grade III | K-L grade IV | Control |
|---|---|---|---|---|
| FTA (°) | 176.8(±3.0) | 180.3(±3.8) | 186.2(±6.7) | 175.6(±3.4) |
| TCA (°) | 11.3(±4.0) | 8.8(±3.9) | 9.8(±3.8) | 9.7(±4.1) |
| TAS (°) | 91.1(±4.7) | 91.4(±4.7) | 90.8(±4.1) | 91.2(±3.4) |
| TTW (°) | 0.4(±1.7) | 0.7(±2.2) | 1.1(±2.3) | 0.3(±1.7) |

Data are presented as mean±standard deviation.

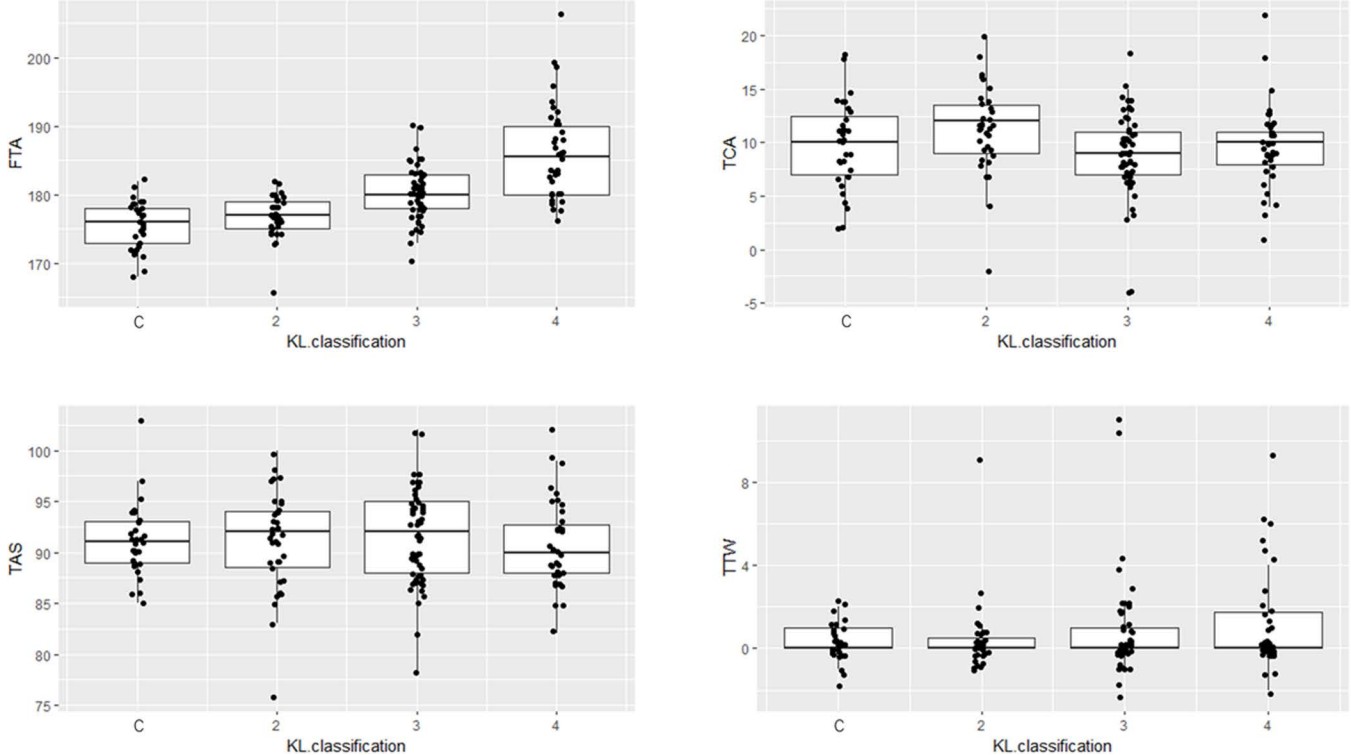

**Fig 3. Significant difference in the angles between each Kellgren–Lawrence classification.** (a) Femorotibial angle (p < 0.05); (b) tibial calcaneal angle (p < 0.05); (c) tibial anterior surface angle (p = 0.94); (d) talar dome during weight-bearing (p = 0.17).

**Table 3. Measurement of grade progression in Kellgren–Lawrence classification and T Takakura–Tanaka classifications.**

|  | K-L grade II | K-L grade III | K-L grade IV | Control |
| --- | --- | --- | --- | --- |
| T-T stage 0 | 14 | 0 | 1 | 31 |
| T-T stage 1 | 17 | 32 | 17 | 0 |
| T-T stage 2 | 3 | 19 | 18 | 0 |
| T-T stage 3 | 1(3a:1) | 3(3b:3) | 2(3a:1, 3b:1) | 0 |
| T-T stage 4 | 0 | 1 | 0 | 0 |

Data are presented as number.

alignment. To assess varus/valgus deformities, we used TAS to confirm the presence and extent of varus/valgus deformities in the talar canal [25,26] and TTW to confirm the presence and extent of varus/valgus deformities in the anterior view of the talofemoral joint [27] in varus-type KOA. To observe hindfoot alignment, we defined the TCA and measured it to determine hindfoot valgus with respect to the tibial axis. As it is possible to identify the contour of the calcaneus in the LL-WB AP view, we decided to measure calcaneal alignment using the contour. Our findings revealed that the hindfoot may compensate for varus-type KOA by changing hindfoot alignment.

In this study, we did not observe a correlation between the TCA and the severity of varus-type KOA. This finding could be attributed to several factors. First, as more than half of the patients in this study had early-stage ankle osteoarthritis (T-T stage 0, 1, 2), the influence of the TAS and TTW on the TCA might have been minimal. Moreover, the varus/valgus of the subtalar joint might have affected the change in the TCA. Although a positive correlation between

knee joint valgus/varus and hindfoot varus/valgus has been reported in KOA using radiographs in the LL-WB AP view [3], no study to date has reported on changes in hindfoot varus/valgus based on KOA severity as per the K–L classification. The present study showed a positive correlation up to grade II of K-L classification of varus-type knee OA and no positive correlation at grade III. The reason may be the limitation or failure of the compensatory function of the hindfoot between grades II and III.

As indicated by the TCA, the observed reduction in medial load, particularly in cases classified as below K–L grade II, may be attributed to the alterations in the TCA shown in our study results. One conservative treatment for KOA is the use of plantar plates. Interestingly, previous reports have shown that lateral wedge insoles with arch support improved knee instability [28] and significantly improved pain and physical function [29]. Particularly, the use of lateral wedge insoles reportedly improves the load values on the medial meniscus and the difference between unloaded and loaded load values, especially in individuals with early varus-type KOA (K–L classification grade II or lower) [30]. A previous report [31] revealed that the lateral wedge plantar plate is effective in verticalizing the load axis from the hip joint to the calcaneus owing to the valgus of the calcaneus at the subtalar joint.

This study had some limitations. First, the calcaneal loading site was not validated in three dimensions. In the present study, we assessed both the knee joint and hindfoot solely from the frontal view of radiographs in the LL-WB AP view. It may be challenging to evaluate the effectiveness of conservative treatment for all severities of knee osteoarthritis using only the two-dimensional evaluation of radiography. We acknowledge the necessity of conducting further research to perform a three-dimensional morphological evaluation of the hindfoot joint surface and identify the point of passage of the lower limb-loading axis. Second, as this was a cross-sectional study, it was not possible to demonstrate a causal relationship. Future longitudinal studies are warranted to address this limitation and elucidate any causal relationships between varus-type KOA severity and hindfoot alignment changes over time.

## Conclusion

In our evaluation using radiographs in the LL-WB AP view, we observed ectropion of the hindfoot in K–L classification grade II, with a subsequent reduction in hindfoot ectropion noted in varus-type KOA grade III cases. These results suggest the potential for alignment correction in the hindfoot and highlight its limitations in the progression of varus-type KOA.

## Acknowledgments

The authors would like to thank Dr. Yuuki Nishimura for his assistance in the research and validation of the measurements.

## Author contributions

**Conceptualization:** Yusuke Ozaki, Ryota Hara, Kensuke Okamura, Hiroaki Kurokawa, Yusuke Inagaki, Munehiro Ogawa, Akira Taniguchi, Yasuhito Tanaka.

**Data curation:** Yusuke Ozaki, Ryota Hara, Yasuhito Tanaka.

**Formal analysis:** Yusuke Ozaki, Ryota Hara, Yasuhito Tanaka.

**Investigation:** Yusuke Ozaki.

**Methodology:** Yusuke Ozaki.

**Writing – original draft:** Yusuke Ozaki, Ryota Hara, Kensuke Okamura, Hiroaki Kurokawa, Yusuke Inagaki, Munehiro Ogawa, Akira Taniguchi, Yasuhito Tanaka.

**Writing – review & editing:** Yusuke Ozaki, Ryota Hara, Kensuke Okamura, Hiroaki Kurokawa, Yusuke Inagaki, Munehiro Ogawa, Akira Taniguchi, Yasuhito Tanaka.

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
