## [Decision Letter · Decision Letter 0]

3 Jul 2024

PONE-D-24-18391Correlation between varus-type knee osteoarthritis severity and hindfoot alignment: analysis of radiographs in the long-leg weight-bearing anteroposterior viewPLOS ONE

Dear Dr. Ozaki,

Thank you for submitting your manuscript to PLOS ONE. After careful consideration, we feel that it has merit but does not fully meet PLOS ONE’s publication criteria as it currently stands. Therefore, we invite you to submit a revised version of the manuscript that addresses the points raised during the review process.

We look forward to receiving your revised manuscript.

Kind regards,

Roham Borazjani

Academic Editor

PLOS ONE

Journal Requirements:

Reviewers' comments:

Reviewer's Responses to Questions

**Comments to the Author**

1. Is the manuscript technically sound, and do the data support the conclusions?

Reviewer #1: Partly

Reviewer #2: Partly

2. Has the statistical analysis been performed appropriately and rigorously? 

Reviewer #1: I Don't Know

Reviewer #2: Yes

3. Have the authors made all data underlying the findings in their manuscript fully available?

Reviewer #1: Yes

Reviewer #2: No

4. Is the manuscript presented in an intelligible fashion and written in standard English?

Reviewer #1: Yes

Reviewer #2: No

5. Review Comments to the Author

Reviewer #1: Thank you very much for allowing me to review your manuscript. I appreciate the effort you have made performing this study and submitting it to plos one. You mentioned that the present study showed a positive correlation up to grade II of K-L classification of varus-type knee OA and no positive correlation at grade III.

What about the correlation at grade 4?

In addition, with your results and conclusion, do you think you can add anything new to the science?

Reviewer #2: This manuscript contains several issues that require correction and clarification:

Abstract:

Rewrite the abstract of the article in a structured format according to the journal's guidelines.

Methodology:

Provide a detailed explanation of the study design in the methodology section.

Clearly state whether the study is retrospective or prospective, case-control or cohort, etc.

Explicitly define the inclusion and exclusion criteria for the study.

Justify why LL-WB AP view technique X-rays were taken in patients with soft tissue inflammation.

Variables:

Cite the studies that introduced the angles variables measured in this study as references.

6. PLOS authors have the option to publish the peer review history of their article (what does this mean? ). If published, this will include your full peer review and any attached files.

**Do you want your identity to be public for this peer review?** For information about this choice, including consent withdrawal, please see our Privacy Policy .

Reviewer #1: No

Reviewer #2: **Yes: ** Sina Afzal

---

## [Author Response · Author response to Decision Letter 1]

25 Sep 2024

1. Is the manuscript technically sound, and do the data support the conclusions?

Author’s answer : Thank you for your comments. In this study, radiographs were taken at the same facility using the same equipment and under the same conditions (please see page 5). The sample sizes was based on the previous study, the references were cited as numbers 3, 4, 5 and 6 (please see page 15). The replication was based on the intraclass correlation coefficients (ICC).

2. Has the statistical analysis been performed appropriately and rigorously?

Author’s answer : Thank you for your comments. In this study, we used One-way analysis of variance (ANOVA) (please see page 8, 9).

3. Have the authors made all data underlying the findings in their manuscript fully available?

Author’s answer : Thank you for your comments. Yes. I have fully disclosed all data. I am attaching the data I used for this study.

4. Is the manuscript presented in an intelligible fashion and written in standard English?

Author’s answer : Thank you for your comments. Yes. I have reviewed and corrected the English grammar of the paper.

5. Review Comments to the Author

Reviewer #1: Thank you very much for allowing me to review your manuscript. I appreciate the effort you have made performing this study and submitting it to plos one. You mentioned that the present study showed a positive correlation up to grade II of K-L classification of varus-type knee OA and no positive correlation at grade III.

What about the correlation at grade 4?

In addition, with your results and conclusion, do you think you can add anything new to the science?

Author’s answer : Thank you for your comments. In K-L classification grade IV, there was again an increase in the angle of TCA, but it cannot be judged as a positive correlation. Based on the results of TCA, we believe that valgus of the sub-talar joint may be compensatory in K-L classification grade II with the progression of KOA varus, and that the compensatory function of the sub-talar joint may not be able to keep up in cases with K-L classification grade III or higher progression.

Reviewer #2: This manuscript contains several issues that require correction and clarification:

Abstract:

Rewrite the abstract of the article in a structured format according to the journal's guidelines.

Author’s answer : Thank you for your comments. I have revised and rewritten the abstract format according to your suggestion.

Methodology:

Provide a detailed explanation of the study design in the methodology section.

Clearly state whether the study is retrospective or prospective, case-control or cohort, etc.

Author’s answer : Thank you for your comments. The study design of this research is a retrospective case-control study.

Explicitly define the inclusion and exclusion criteria for the study.

Justify why LL-WB AP view technique X-rays were taken in patients with soft tissue inflammation.

Author’s answer : Thank you for your comments. Inclusion and exclusion criteria for the study are described in the first paragraph of “Study population where you specify inclusion and exclusion criteria” (please see page 5).

The reason for performing LL-WB AP View in patients with soft tissue diseases such as anterior cruciate ligament injuries was to measure the severity of knee deformity in K-L Grade-0 patients using the same imaging method as other Grade patients.

Variables:

Cite the studies that introduced the angles variables measured in this study as references.

Author’s answer : Thank you for your comments. The references of FTA were cited as numbers 1, 23 and 24 (please see page 15, 19). The references of TAS were cited as numbers 25 and 26 (please see page 19). The references of TTW were cited as numbers 27 (please see page 19). The TCA definition and angle measurement are original angles that exist only in this study as far as I have been able to determine.

6. PLOS authors have the option to publish the peer review history of their article (what does this mean?). If published, this will include your full peer review and any attached files.

Do you want your identity to be public for this peer review? For information about this choice, including consent withdrawal, please see our Privacy Policy.

Author’s answer : Thank you for your comments. Yes.

---

## [Decision Letter · Decision Letter 1]

5 Feb 2025

PONE-D-24-18391R1Correlation between varus-type knee osteoarthritis severity and hindfoot alignment: analysis of radiographs in the long-leg weight-bearing anteroposterior viewPLOS ONE

Dear Dr. Ozaki,

Thank you for submitting your manuscript to PLOS ONE. After careful consideration, we feel that it has merit but does not fully meet PLOS ONE’s publication criteria as it currently stands. Therefore, we invite you to submit a revised version of the manuscript that addresses the points raised during the review process.

We look forward to receiving your revised manuscript.

Kind regards,

Xindie Zhou

Academic Editor

PLOS ONE

Journal Requirements:

Reviewers' comments:

Reviewer's Responses to Questions

**Comments to the Author**

1. If the authors have adequately addressed your comments raised in a previous round of review and you feel that this manuscript is now acceptable for publication, you may indicate that here to bypass the “Comments to the Author” section, enter your conflict of interest statement in the “Confidential to Editor” section, and submit your "Accept" recommendation.

Reviewer #1: (No Response)

2. Is the manuscript technically sound, and do the data support the conclusions?

Reviewer #1: Yes

3. Has the statistical analysis been performed appropriately and rigorously? 

Reviewer #1: I Don't Know

4. Have the authors made all data underlying the findings in their manuscript fully available?

Reviewer #1: Yes

5. Is the manuscript presented in an intelligible fashion and written in standard English?

Reviewer #1: Yes

6. Review Comments to the Author

Reviewer #1: Thank you very much for allowing me to review your manuscript. I appreciate the effort you have made performing this study and submitting it to plos one.

I think the authors have to mark the changes they made for revision with different color.

In addition, they did not respond to this question “ with your results and conclusion, do you think you can add anything new to the science?”

7. PLOS authors have the option to publish the peer review history of their article (what does this mean? ). If published, this will include your full peer review and any attached files.

**Do you want your identity to be public for this peer review?** For information about this choice, including consent withdrawal, please see our Privacy Policy .

Reviewer #1: No

---

## [Author Response · Author response to Decision Letter 2]

6 Apr 2025

Response to reviewer comments

Dear Reviewers,

We would like to thank the reviewers for your time and efforts in reviewing our manuscript and for providing comments, which have considerably helped us improve our manuscript. We have made revisions based on your comments and have provided our point-by-point responses below. We hope that our responses and revisions appropriately address your comments.

3. Has the statistical analysis been performed appropriately and rigorously?

Reviewer #3: I Don't Know.

Author #3: Thank you very much for allowing me to review your manuscript. I used ANOVA to assess differences between the stage and control groups in the T–T classification at the TCA, and Post-hoc analysis was performed using the Bonferroni test for multiple comparisons (please to see page 8). Based on the results of the analysis of variance, the probability of a significant difference between grade 2 and grade 3 of the K-L classification was determined to be 0.05 or less, and a significant difference was recognised. The results of the analysis of variance for TCA between each K-L classification are attached.

The variables are as follows: variable = TCA, independent variables = age/height/weight/BMI/K-L classification

K-Lgrade 0+1/2

sum of spuares degree of freedom mean spuares F-measure significance level

regression 109.798 6 18.3 1.086 0.381

residual 994.156 59 16.85

total 1103.955 65

K-Lgrade 2/3

sum of spuares degree of freedom mean spuares F-measure significance level

regression 228.129 6 38.022 2.422 0.033

residual 1302.86 83 15.697

total 1530.989 89

K-Lgrade 3/4

sum of spuares degree of freedom mean spuares F-measure significance level

regression 69.041 6 11.507 0.732 0.625

residual 1352.078 86 15.722

total 1421.118 92

6. Review Comments to the Author

Reviewer #6: Please use the space provided to explain your answers to the questions above. You may also include additional comments for the author, including concerns about dual publication, research ethics, or publication ethics. (Please upload your review as an attachment if it exceeds 20,000 characters)

Author #6: First of all, thank you for pointing out the corrections. I have made the corrections by adding different colors.

Next, I will answer your question. My answer is “By studying the effects of knee osteoarthritis on the talocrural joint and sub-talar joint, I believe that it has become even clearer than in previous reports that it is possible to distinguish between the limit of conservative treatment for osteoarthritis and the timing of surgical treatment intervention in order to preserve the hindfoot. This indicates the appropriate timing for both conservative and surgical treatment, and I believe that it shows the possibility of providing more appropriate treatment by reducing the disadvantages and unnecessary physical and financial burdens on patients.”.

Do you want your identity to be public for this peer review? For information about this choice, including consent withdrawal, please see our Privacy Policy.

Author response: Yes.

Best regards

Yusuke Ozaki

Orthopaedic Department, Nara Medical.University, Kashihara city, Nara prefecture, Japan

---

## [Editor Report · Decision Letter 2]

5 May 2025

Correlation between varus-type knee osteoarthritis severity and hindfoot alignment: analysis of radiographs in the long-leg weight-bearing anteroposterior view

PONE-D-24-18391R2

Dear Dr. Ozaki,

We’re pleased to inform you that your manuscript has been judged scientifically suitable for publication and will be formally accepted for publication once it meets all outstanding technical requirements.

Kind regards,

Xindie Zhou

Academic Editor

PLOS ONE
---

## [Editor Report · Acceptance letter]

PONE-D-24-18391R2

PLOS ONE

Dear Dr. Ozaki,

I'm pleased to inform you that your manuscript has been deemed suitable for publication in PLOS ONE. Congratulations! Your manuscript is now being handed over to our production team.

Kind regards,

on behalf of

Dr. Xindie Zhou

Academic Editor

PLOS ONE